# Physicochemical Characterization and Evaluation of Emulsions Containing Chemically Modified Fats and Different Hydrocolloids

**DOI:** 10.3390/biom10010115

**Published:** 2020-01-09

**Authors:** Małgorzata Kowalska, Magdalena Woźniak, Anna Żbikowska, Mariola Kozłowska

**Affiliations:** 1Department of Chemistry and Organic Materials, Faculty of Chemical Engineering and Commodity Science, Kazimierz Pulaski University of Technology and Humanities, 27 Chrobrego St, 26-600 Radom, Poland; 2Department of Food Technology, Faculty of Food Sciences, Warsaw University of Life Sciences—SGGW, Nowoursynowska 159C, 02-787 Warsaw, Poland; anna_zbikowska@sggw.pl; 3Department of Chemistry, Institute of Food Sciences, Warsaw University of Life Sciences—SGGW, Nowoursynowska 159C, 02-787 Warsaw, Poland; mariola_kozlowska@sggw.pl

**Keywords:** lipids, structured fats, calf tallow, pumpkin oil, emulsion, thickeners

## Abstract

The study aims to investigate the physicochemical properties and stability of the dispersion systems containing structured fats as a fatty base. In this work, calf tallow and pumpkin seed oil blends were chemically interesterified at various ratios (9:1, 3:1, 3:2, 3:3, 2:3, and 1:3) to produce structured lipids. Fatty acids composition, polar and nonpolar fraction content, and acid value were determined for the raw fats and interesterified blends. Afterwards, selected blends were applied in emulsion systems. Stability, microstructure, color and texture of emulsions were evaluated. The chemical interesterification had an effect on the modified blends properties, and caused an increase in polar fraction content and acid value, and a decrease in nonpolar fraction content. No effect on the fatty acids composition has been found. The evaluation of the prepared emulsions results allowed us to select two of the most stable and favorable samples—both containing chemically interesterified calf tallow and a pumpkin seed oil blend in a ratio of 1:3 as a fatty base, and xanthan gum or carboxymethylcellulose as a thickener. The obtained dispersions, containing fatty bases with improved physicochemical properties and desirable functionality, can be applied as food, cosmetic, and pharmaceutical emulsions.

## 1. Introduction

Lipids represent a major class of biomolecules, including a wide range of chemical structures. These compounds fulfill a variety of roles in living organisms as, among others, energy storage to fuel metabolism; they are structural components and participate in signal transduction [1]. Native lipids of vegetable of animal origin do not always fulfill the requirements or needs. Achieving the lipid products with enhanced nutritional value while keeping physical properties within the desired range may constitute a scientific challenge. The physical characteristics of vegetable oils and animal fats are determined by chain length and degree of unsaturation in the fatty acids (FA), as well as their distribution in the glycerol backbone [2]. Among these properties, the solid fat content is of great importance since physicochemical properties and sensory attributes of many products are determined by the fraction of molecules (triacylglycerols) that are solidified at a given temperature. Examples include the food and cosmetics industries, in which the melting profile affects final-product sensory features such as texture, creaminess, hardness, taste, and spreadability [3].

Fat modifications are frequently proposed for the development of more healthy lipid products. Usually, the modifications have a significant impact on the crystallization behavior, and thus on the structure of fat-containing final products [4]. Chemical interesterification (CIE) is an important technological option for the production of fats targeting commercial applications [5]. Structured lipids, which can be obtained by chemical interesterification, are triacylglycerols (TAG) that have been modified by the incorporation of new FA in a glycerol backbone, change of FA positions, or altered to yield novel TAG not found in nature. Therefore, CIE reaction modifies the physical characteristics of fats and oils by rearrangement of the FA distribution on the glycerol backbone, nevertheless without changing their chemical composition [6]. Generally, blends of hard fats and liquid oils through the interesterification process are modified to meet the desired physicochemical properties [7]. Such modification is expected to improve the desirable characteristics of the lipid blends, and it allows the production of fats with higher plasticity, suitable melting point, and with a desired fatty acid composition [8]. Moreover, beneficial from the food industry’s point of view, both chemical and enzymatic interesterification reactions are important methods due to the fact that they do not contribute to trans fatty acid formation [5,9,10,11].

The research concerning new blends of animal fats and vegetable oils, demonstrating desired and enhanced nutritional properties, is usually based on prior experience and direct experimental studies and is widely reported on literature [2,12,13,14]. According to the best of our knowledge, despite the high number of these valuable experimental works, the application of fats modified via chemical interesterification as a fatty base of cosmetic emulsions is still deficient. In the study, we used calf tallow and pumpkin seed oil as components of interesterified blends. Beef and calf tallow, are a by-product of the meat industry and generally are considered as not suitable for direct human consumption due to, among others, low nutritional value [15]. However, these fats have beneficial properties such as high thermal and oxidative stability and favorable plasticity at room temperature [16]. Triacylglycerols of these fats contain a relatively high content of saturated fatty acids, which can cause a sandy sensation. Under conditions of frequent temperature changes (e.g., during storage or transport), the crystal structure of this fat may change, leading to the formation of crystals up to a few millimeters in size [16]. The choice of pumpkin seed oil in our research was dictated by, among others, high content of bioactive compounds as tocopherols, sterols, β-carotene, and lutein [17]. The high content of unsaturated fatty acids in this oil makes it well-suited for improving nutritional benefits [18]. Due to the characteristic nutty taste as well as deep color, it could be a novelty for model fat emulsions.

Since emulsions represent a mixture of at least two immiscible liquids, they are, according to the law of thermodynamics, inherently unstable. These systems require the addition of suitable stabilizers to guarantee a long shelf life. Generally, ionic or non-ionic surfactants are used as emulsifiers and stabilizers in the form of hydrocolloids. Hydrocolloids are high-molecular-weight hydrophilic biopolymers used as functional ingredients in the food and cosmetic industry to control the structure, texture, and to form a required consistency of a final product. The term “hydrocolloid” refers to the majority of polysaccharides extracted from plants, seaweeds, and microbial sources as well as gums derived from plant exudates, and modified biopolymers manufactured by chemical or enzymatic treatment of starch or cellulose [19]. These compounds influence the properties of dispersed systems due to their interfacial properties [20]. Surface-active hydrocolloids may act as emulsifiers and stabilizers through adsorption of a protective layer at the oil–water interface. Interactions of hydrocolloids with emulsion-dispersed phase droplets may affect the rheology and stability with respect to aggregation and serum separation. Another type of emulsifier is lecithins, which can also be used as viscosity regulators and dispersing agents. They consist of a blend of various phospholipids with other substances such as fatty acids, triacylglycerols, and carbohydrates [21]. Lecithins are widely used for food, cosmetic, and pharmaceutical applications. In the study, sunflower lecithin was used, which revealed good emulsifying properties and prevented destabilizing processes in emulsions.

The requirements for thickeners are continuously increasing, which is associated with the demand for obtaining the required stability and product purpose, and taking into account the requirements of consumers. The thickeners used in the study (maltodextrin, microcrystalline cellulose and xanthan gum, xanthan gum, carboxymethylcellulose) were chosen on the basis of literature study and earlier-conducted studies on the properties of various hydrocolloids. In this presented work, we propose to use chemically modified (interesterified) lipid blends of vegetable oil with animal fat to achieve products with enhanced nutritional value while keeping physical properties within a desired range. Also, the objective of this framework is to select an emulsion containing modified fats and various thickeners and to reveal the highest stability. Generally, the present study seeks to develop structured lipids based on pumpkin seed oil and calf tallow blends at various ratios aiming to obtain a product with better physical and chemical properties and desirable functionality to be applied in food, cosmetic, or pharmaceutical emulsions.

## 2. Materials and Methods

### 2.1. Material

Calf tallow (CT) was obtained from a local farm near Radom, Poland. Pumpkin seed oil (PO) was obtained from Oleofarm (Wrocław, Poland). Reagents used were sodium methoxide (Sigma Aldrich, Steinheim, Germany), phosphoric acid (Avantor, Gliwice, Poland), diethyl ether (Avantor, Gliwice, Poland), magnesium sulfate (Chempur, Piekary Ślaskie, Poland), heptane (Chempur, Piekary Ślaskie, Poland), sodium hydroxide (Acros Organics, Geel, Belgium), and methanol (Acros Organics, Belgium). For emulsions preparation following components were used: sunflower lecithin (Lasenor, Barcelona, Spain), maltodextrin (Hortimex, Konin, Poland), Vivapur CS 032 XV (J. Rettenmaier & Sohne, Rosenberg, Germany)-microcrystalline cellulose and xanthan gum blend, xanthan gum (Brenntag, Kędzierzyn Koźle, Poland), carboxymethylcellulose (Barentz, Hoofddorp, Netherlands), and sodium benzoate (Brenntag, Kędzierzyn Koźle, Poland).

### 2.2. Methods for Fats Preparation and Analysis

#### 2.2.1. Calf Tallow Bleaching

Calf tallow was bleached and deodorized before the reaction. The tallow was melted, placed in a two-neck round-bottom flask, and 2 wt% of bleaching earth was added. The blend was heated under a reflux condenser at 80 °C for 1 h. Then, the sorbent was filtered at 70 °C using a paper filter.

#### 2.2.2. Chemical Interesterification (CIE)

CF and PO were mixed in different ratios (9:1, 3:1, 3:2, 3:3, 2:3, and 1:3). The closed Erlenmeyer flasks containing the fat blends were placed in a water bath with shaking (SWB 22N, Labo Play, Poland) and thermostated at 90 °C for 15 min. Afterwards, chemical catalyst—sodium methoxide was added in the amount of 0.6 wt%. After 2 h, reaction was stopped by the addition of 50 cm^3^ of a diluted and heated solution of H_3_PO_4_ to neutralize the catalyst. After cooling to ambient temperature, fats were extracted with diethyl ether (3 × 80 mL) and separated from the aqueous phase. The organic phase was dried with magnesium sulphate, which was then filtered. The solvent was evaporated.

#### 2.2.3. Acid Value (AV) of Fats

The AV was determined according to ISO method (PN EN ISO 660, 2009) [22], based on titration with 0.1 M aqueous solution of potassium hydroxide using phenolphthalein as an indicator. Fat samples were dissolved in a mixture of equivalent amounts (*v/v*) of diethyl ether and ethanol. The content of FFA (% FFA) was calculated on the basis of the AV. The results were presented as a mean of 3 determinations ± standard deviation (SD).

#### 2.2.4. Composition of Fats

The percentage content of TAG, DAG, and MAG was determined using gel permeation chromatography (GPC) by means of an Agilent 1100 series. Samples were dissolved in THF to obtain a final concentration of 0.5%. Isocratic elution (flow: 1 mL/min) with tetrahydrofuran, stabilized with BHT, was used on two Phenogel columns (Phenomenex, Izabelin-Warsaw, Poland 300 × 7.80 mm, 5 micron, 100 Å and Phenomenex, 300 × 7.80 mm, 5 micron, 50 Å). Detection with refractive index was used, calculations were made on the basis of the calibration curves. The measurements were carried out in triplicate and were presented as a mean ± standard deviation (SD).

#### 2.2.5. Fatty Acids Profile

Analysis of fatty acid methyl esters (formed via fats esterification reaction) was performed with an Agilent Technologies 6890N gas chromatograph (Agilent Technologies, Wilmington, DE, USA), equipped with a flame-ionization detector and an Alltech EC-Wax capillary column (30 m, 0.25 mm ID, 0.25 µm film thickness). The oven temperature ramp program was 50–180 °C at 10 °C/min, then increased to 240 °C at 5 °C/min. The detector and injector were set at 300 and 250 °C, respectively. The splitless mode was used. All percentage values determined by GC for each fatty acid methyl ester were the mean of triplicate runs.

### 2.3. Methods for Emulsions Preparation and Analysis

#### 2.3.1. Emulsion Preparation

The oil phase of all emulsions consisted of chemically interesterified fat blends: CT:PO 1:3, CT:PO 3:1, and lecithin (Table 1). The aqueous phase of each emulsion was prepared by dispersing an appropriate thickener provided in Table 1 in distilled water over 1 min. Afterwards, both phases were heated to 50–55 °C. Then, the phases were mixed and homogenized for 4 min at 18500 rpm by means of a T18 digital ULTRA-TURRAX homogenizer equipped with an S18G-19G dispersing head (IKA, Shanghai, China). After cooling the emulsions to ambient temperature, a preservative—sodium benzoate—was added. The percentage composition of the components was based on our previous research [23,24].

#### 2.3.2. Microphotographs of Emulsions

Microphotographs of the freshly prepared emulsions were taken using an optical microscope (Genetic Pro Trino, Delta Optical, Warszawa, Poland) and a digital camera (DLT Cam Pro, Delta Optical, Poland) at a total magnification of 400×.

#### 2.3.3. Colour Determination of Emulsions

Colour determination was performed using a Konica Minolta chromameter CR-400 (Konica Minolta Sensing Inc., Milton Keynes, UK) after standardization with a white calibration plate. CIEL*a*b* system was used with the following values: L*—defined as the lightness of the sample ranging from 0 (black) to 100 (white), a* and b* represents two perpendicular color axes, with the values ranging from −60 to +60. Parameter a* when (−) represents greenness, when (+) represents redness. Whereas b*, blueness when (−) and yellowness when (+) [25]. The measurements were taken on stored emulsions (30 days, 5°C) in triplicate and are presented as a mean ± standard deviation (SD).

#### 2.3.4. Viscosity Determination of Emulsions

The viscosity of the emulsions was determined using a Brookfield DV-III Ultra rheometer, model HA with a helipath spindle set (Brookfield Engineering Laboratories, Commerce Blvd Middleboro, MA USA). Measurements were carried out at a constant rotational spindle speed of 10 rpm using T-bar spindle no. 92 (T-B). The measurements were performed at 20 °C in triplicate and are presented as a mean ± standard deviation (SD).

#### 2.3.5. Turbiscan Test

The kinetic of emulsions destabilization was assessed by means of a Turbiscan Lab (Formulation, Toulouze, France). The device uses a pulsed near-IR light source at a wavelength of 880 nm. Two synchronized detectors collect light transmitted (T) and backscattered (BS) by a sample. Variation of the BS and T intensities are caused by changes in the droplet volume fraction migration or mean particle size [26]. The measurements were performed at a temperature of 25 °C on samples stored at 30 °C for a few days interval. The data was presented as a percentage variation of ΔBS as a function of time and sample height (the curves obtained by subtracting the BS profile at t = 0 from the profile at t (ΔBS = BS_t_ − BS_0_)). TSI (the Turbiscan stability index) was calculated on a basis of the following formula:(1)$$TSI=\underset{i}{\sum }\frac{\Sigma_{h}\left|scan_{i}\left(h\right)-scan_{i-1}\left(h\right)\right|}{H}$$
where *scan_i_*(*h*) is mean BS for each *i* of measurement, *scan_i−_*_1_ (*h*) is mean BS for the *i* − 1 measurement, and *H* is the height of a sample [27]. Higher TSI indicates stronger destabilization caused by particle aggregation and dynamic migration. The above-mentioned calculations were carried out using TurbiSoft 2.0. software.

#### 2.3.6. Droplet Size of Emulsions

By means of Turbiscan Lab (Formulation, France), emulsion droplet diameter was determined. For this purpose, the dependence of BS intensity variations on the size of the mean droplet diameter (D) and particle volume fraction (ø), i.e., BS = f (D, ø), was used [28]. The calculations were carried out using TurbiSoft 2.0. software.

#### 2.3.7. Texture Analysis of Emulsions

Texture determination was performed using a Texture Analyzer CT3 (Brookfield Engineering Laboratories, Inc., Middleboro, MA, USA) equipped with Brookfield Texture Pro CT software. One-cycle compression mode was used. As a probe, a nylon spherical-shaped probe (TA43; diameter 25.4 mm) was used. All samples were placed in the same container type (cylindrical: 70 mm depth; 50 mm internal diameter). The test parameters were as follows: test and return speed 0.5 mm/s, pre-test speed 2 mm/s, target depth 10 mm, trigger load 1 g, data rate 10 points/s. The measurements were carried out on freshly prepared as well as stored emulsions (30 days, 5 °C) in triplicate at room temperature.

#### 2.3.8. Statistical Analysis

A one-way analysis of variance and Tukey’s test were used to establish the significance of differences between the means (*p* < 0.05). The statistical analysis was carried out with the Statgraphics plus 4.0 package (Statistical Graphics Corp., The Plains, VA, USA).

## 3. Results and Discussion

Physicochemical as well as nutritional properties of the natural lipids depend on the structure and composition of triacylglycerols [29]. These properties are not always mutually compatible. Thus, fat modifications can contribute to give them specific functionalities, increase their oxidative stability, or even improve their nutritional value (increase in unsaturated fatty acids content). Thus, chemical interesterification reaction may be used to produce structured lipids and contribute to improving their properties.

Fatty acid composition is one of the parameters determining the quality of hard fats and oils. In the study, it was found that the composition of fatty acids was various and depended on the amount of animal fat and oil in the fat blend. In general, interesterification did not have a major impact on the content of fatty acids in the blends, and the variations in their content was mainly related to the original composition of the blended fats.

The fatty acids (FA) composition of calf tallow, pumpkin seed oil, and their chemically interesterified blends is given in Table 2. Pumpkin seed oil contained a high amount of linoleic acid (46.7%), while the major fatty acid in calf tallow was oleic acid, which was present at 35.4%. Oleic acid content was comparable to the amount present in pumpkin seed oil (31.9%). Pumpkin seed oil contained a significantly smaller amount of palmitic acid and nearly half less stearic acid than calf tallow. Nevertheless, an increase in the amount of palmitic acid, in relation to the amount of this acid in pumpkin seed oil, was observed in all newly formed fat blends. According to Jeyarani and Reddy [30], higher palmitic acid content increases tendency to the β′ crystal formation, which results in a desirable smooth consistency of fats. Generally, after the interesterification reaction, trans isomers formation was not observed in the new fat blends. Asif [31] claims that, from the nutritional point of view, interesterification contributes to the elimination or reduction of trans fatty acids, providing higher essential fatty acid activity. In our research, this situation was observed for the modified fat blend for which the content of pumpkin seed oil was the highest (75%). This structured lipid had the highest content of unsaturated fatty acids, and the main fatty acids were linoleic and oleic acid (Table 2). No effect on the fatty acids composition has been found. Oliveira et al. [14] reported similar results.

The content and position of individual fatty acids in the triacylglycerol molecules are major properties of the fats, which have an effect on the consistency as well as the viscosity of the obtained lipids. An important parameter influencing the quality of the obtained fat blends is the content of the polar and nonpolar fractions. The study showed that the highest amount of triacylglycerol fraction was found in the blend containing the highest share of pumpkin seed oil (CT:PO 1:3), while the least was noted for the blend containing the highest share of calf tallow (CT:PO 9:1). For the CT:PO 3:1 and CT:PO 3:2 blends also containing a higher share of the animal fat, a certain amount of monoacylglycerols was formed after the interesterification reaction. For the remaining blends, a small amount of these compounds was formed, which indicates a different time to complete the reaction. After the interesterification process, an increase in free fatty acids content was observed for all blends. These results are consistent with the results of Amir et al. [32], who indicated that the interesterification process generates greater amounts of free fatty acids in the products obtained. The most favorable low amount of free fatty acids was obtained for the CT:PO 1:3 blend. According to Hu and Jacobsen [33], the presence of free fatty acids accelerates adverse processes affecting the quality of fats, i.e., they contribute to rancidity.

Generally, the interesterification process influenced the change of the polar and nonpolar fractions content. A decrease in the content of triacylglycerols was observed and an increase in monoglycerols, diacylglycerols, and free fatty acid content in relation to the raw fats (Table 3). These data are consistent with the observations indicated by Bryś et al. [34] who obtained an increase in the polar fraction content with a simultaneous decrease in the triacylglycerol fraction after the interesterification reaction. However, increased content of monoacylglycerols and diacylglycerols can contribute to a favorable fatty base when applied in emulsions, showing emulsifying properties [35]. Monoacylglycerols and diacylglycerols are compounds that are used as emulsifiers and substances stabilizing emulsion systems [36].

Figure 1 shows acid values (AV) of the raw fats and interesterified fat blends. AV of the initial fats was 0.8 and 1.7 for calf tallow for pumpkin seed oil, respectively. After the interesterification reaction, the acid value for the interesterified blends was in a range of 2.4–17.4. It was observed that the AV varied, and generally increased, with an increase of the hard fat content in the lipid blend. This means that a greater proportion of animal fat in the modified fat mixture had an effect on more advanced fat hydrolysis—a process that can occur in parallel during interesterification. According to the authors [8], the increase in the value of the acid number can also be a confirmation of the occurring changes in fat, i.e., its modification. The above results indicate that the safest fat in terms of possible adverse changes generated by the presence of free fatty acids is fat blend CT:PO 1:3. A different observation was made by Seriburi and Akoh [37], who performed an enzymatic interesterification of lard and high oleic sunflower oil. With an increasing amount of lard, a decreasing amount of free fatty acids was noted. The observed increase tendency for this parameter after a interesterification reaction is also consistent with the results of other researchers [14,29,38].

Petrauskaite et al. [39] and Oliveira et al. [14] also confirm that an increase of AV after interesterification proves good catalyst activity. Oliveira et al. [14] also reported that the decrease in FFA content in the fat blend during interesterification can be a result of an interaction of catalyst with fat acidity. For fats with high FFA content, sodium methoxide neutralizes the free fatty acids and consequently, their content decreases after the interesterification. Thus, it results in greater usage of a catalyst, which is required to neutralize FFA present in the reaction environment [7].

In the next part of this study, from all six interesterified fat blends, two of them (CT:PO 3:1 and CT:PO 1:3) were selected and used as the fatty base of the emulsions. The choice of these fat blends was dictated by different plasticity and consistency, which, according to the authors, could also have an impact on their stability. Further research aimed to determine the quality of emulsions prepared by determining their chosen physicochemical parameters.

Emulsions stability evaluated on a basis of the visual appearance was high for all the prepared samples (Figure 2). There was no phase separation observed for the emulsions formed with various thickeners after 24 h from their manufacturing as well as after a 30-day storage period. Although, a small non-significant amount of oil droplets was observed on the surface of emulsion E7 on a first day of storage, nevertheless no sedimentation or cream layer was observed visually during the storage period.

Emulsion stability is an ability to resist variations in physicochemical properties during storage [40]. Thus, it can be concluded that it is a measure of the rate at which an emulsion creams, sediments, coalesces, or flocculates. The rate of these changes can be measured by determination of the emulsion’s droplet size and their distribution, or by monitoring of the changes using Turbiscan analysis. Droplet measurement is particularly important since it determines many of the most significant properties of emulsion-based products (e.g., shelf life, appearance, and texture and flavor).

The average particle size of all the prepared emulsion systems ranged from 3.2 to 11.8 µm. After the storage period, emulsions E5 and E6 showed the highest particle size (Figure 3). This results suggested that there was an insufficient amount of a thickener to cover the oil droplets, which led to droplet flocculation and, thus, increased their size. Similar conclusions have also been made by Goyal et al. [41] and Ma et al. [42], who determined the insufficient level of a thickener in obtained systems. The instability leading to a significant increase in the average particle size may also be related to the effectiveness of a thickener or can be associated with the type of fat blend applied as a fatty base in an emulsion. A significant droplet size increase was observed for emulsions containing a fatty base with a predominant share of calf tallow, and for which the thickeners were MD and MC&XG. The droplet size of the emulsions E3 and E4 remained practically unchanged after a 30-day storage period (an increase of 0.1 and 0.2, respectively; Figure 3).

The Turbiscan test, based on transmitted (T) and backscattered (BS) light intensity, was used to analyze the stability of the emulsions based on modified fats. Figure 4 shows backscattered light intensity profiles of the prepared emulsions. The backscattered light intensity is related to the stability of an emulsion—more precisely to the physical processes occurring during storage. The profiles are presented in a reference mode, which means that the initial scan is presented as a baseline (ΔBS = 0%). The variation of the BS light intensity is correlated to the droplets’ migration or their size variations.

In general, the highest variations of the BS light intensity were observed for the emulsions containing the fatty base with a greater share of calf tallow (CT:PO 3:1). For emulsion E8, the ΔBS signal decreased at the top of the sample due to a decrease of suspension concentration. The peak thickness gradually increased following the time. Such changes may indicate a predisposition of a system to possible destabilization changes of a gravitational type [43]. However, the greatest changes in the average particle size, as evidenced by the non-overlapping line in the middle part of the graphs, as well as the changes in the type of creaming, were observed for the emulsions E5 and E6.

The ΔBS profiles of the emulsions E1–E4, containing a fatty base with a greater share of pumpkin seed oil (CT:PO 1:3), were similar. In the middle part of the graphs, the curves were rather overlapped, suggesting a slight variation of the droplet size for these emulsions over time (Figure 4). Lorenzo et al. [44] have studied the effect of xanthan gum as a fat substitute on emulsion stability. According to the authors, incorporation of a xanthan gum is a suitable alternative to stabilize low-in-fat o/w emulsions against creaming, and the visual inspection of the investigated emulsions showed that they remained stable after eight months of storage. The above observations are consistent with the results of the obtained TSI results. The lowest values of this parameter, which were practically unchanged over the storage period, were noted for the emulsions E3 and E4. In contrast, the greatest increase in TSI values over time was recorded for emulsions E5 and E6 (Figure 5), which confirms the significant destabilization changes in these systems.

Figure 6 shows the microstructure of the emulsion observed by means of an optical microscope. The correct dispersion of fat droplets was observed for each emulsion sample, which confirmed the lack of phase separation of the systems. The most favorable dispersion indicating homogeneity and small particle size was found to be the emulsion E4. The information obtained during this measurement was consistent with the data obtained from the measurement of the average particle size which indicated the lowest particle for this emulsion. A similar dispersion microstructure was observed for the emulsion E8. This fact may indicate that emulsions containing CMC as a thickener were properly dispersed. Whereas, other factors, such as the type of an oil phase in the case of our study, may affect the quality of an emulsion during storage for a certain period of time. It was also observed that emulsions containing MD (E1, E5) and MC&XG (E2, E6) as the thickening agents were characterized by greater droplets.

Another parameter which was evaluated was the emulsions’ viscosity. The viscosity values were varied and depended on type of the thickener and the concentration of vegetable oil in a chemically modified fatty base. An increase in viscosity for each emulsion after the storage was observed. A significantly greater difference in viscosity values after the 30-days storage period was noted for emulsions containing chemically interesterified fat blends in a ratio of CT:PO 3:1 than in a ratio of CT:PO 1:3. In our study, the fat content was the same in all the prepared emulsions, although the difference in the composition of the fat blends used as fatty bases of emulsions was significant. The amount of animal fat present in the fat blend affected the rheological properties of emulsion systems. Comparing the type of thickener used, it was found that for both fat blends used as the fatty bases, the difference in viscosity values was the smallest after the storage period, when the mixture of MC & XG was used (902 mPas). The largest changes, and at the same time the largest increase in viscosity, was noted for the emulsion E7 stabilized with XG (11 153 mPas). The highest viscosity among all the emulsions after 24 h was noted for emulsion stabilized with an MC & XG, and the lowest for XG stabilized emulsion. However, the smallest differences in the values of the viscosity parameter during the entire storage period were observed for the emulsions E1 and E2 (Figure 7).

Analyzing the texture properties of the emulsions with specific thickeners, it was noted that the freshly prepared samples containing CT:PO 1:3 as a fatty base were characterized by lower hardness values than their counterparts containing CT:PO 3:1 (Figure 8). It was probably caused by the emulsions’ oil base fatty acids composition. For the emulsions containing greater share of animal fat, a significantly higher content of saturated FA was noted, which have an impact on higher hardness values of the emulsions. Considering the evolution of this parameter values after the 30-days storage period, it was observed that for emulsions containing MD and MC & XG (E1, E2, E4, and E5) as a thickeners, an increase in this parameter was observed. The opposite observation was found for emulsions containing XG and CMC (E1, E2, E4, and E5), the decrease in the hardness values was noted. The highest hardness values for the freshly prepared as well as stored sample, among all tested emulsions, was found for emulsion E7 (50 and 43 g, respectively).

Sample adhesiveness can be identified as its stickiness. There was a general tendency observed that emulsions containing the predominant share of calf tallow in the modified fatty base showed greater values of adhesiveness; thus, can be considered more sticky than the ones that contained a greater share of pumpkin seed oil. The same observation, as for hardness values, was found. The decrease or increase in this parameter after the storage period depended on the thickener type.

The color change of emulsion products is one of the important quality determinants that can inform about processes occurring in the product and can indicate destabilization changes [40]. Analyzing the L* parameter values indicating sample brightness, it was found that emulsions with higher share of pumpkin seed oil were characterized by a slightly darker shade of color than the emulsion for which the fatty base was fat blend containing 75% of animal fat (Table 4). After storage, it was found that the color of the emulsions became darker with both fatty bases used. After a 30-day storage period, the a* parameter showed slightly higher negative values for all the prepared emulsions, which indicates a change in color towards the green shades. The decrease in the b* parameter value after the storage period indicated that the color of the emulsion tended to change towards blue shades. However, the above changes observed cannot be counted as clear changes indicating advanced destabilization changes. Rather, these are changes captured due to the sensitivity of the device and, in the authors’ opinion, do not indicate lower quality of the systems after the storage period. Also, visual analysis of the emulsion did not confirm distinct noticeable changes indicating the destabilization of systems. No clusters of color, migration of particles and their accumulation were observed in specific parts of the vials in which they were stored (Figure 2).

## 4. Conclusions

The interesterification reaction affected significantly the modified blends properties, and caused an increase in polar fraction content and acid value, and a decrease in nonpolar fraction content. No effect on the fatty acid composition was found. The evaluation of the prepared emulsions results allowed us to select two of the most stable and favorable samples—both containing chemically interesterified CT:PO 1:3 blend as a fatty base, and XG or CMC as a thickeners.

The method proposed in the work promotes the use of meat industry by-product (calf tallow), while pointing to its valuable rheological features. The study revealed that chemically interesterified waste calf tallow and pumpkin seed oil blends can be well-suited for obtaining a stable model emulsion system. In our opinion, the emulsions can be used in several industries, such as food, cosmetics, or even pharmaceutics. The resulting new products are in line with the principles of sustainable development, as well as with current trends in the use of natural raw materials. The benefit of this modification is the production of new fat enriched with unsaturated fatty acids which, being a component of food emulsions, will reveal beneficial nutritional properties, and while being a component of cosmetic emulsion, will be responsible for proper hydration and skin condition. The use of pumpkin seed oil, rich in unsaturated fatty acids, makes a final product attractive, not only from the nutritional point of view but also in a possible application on skin.

## Figures and Tables

**Figure 1 biomolecules-10-00115-f001:**
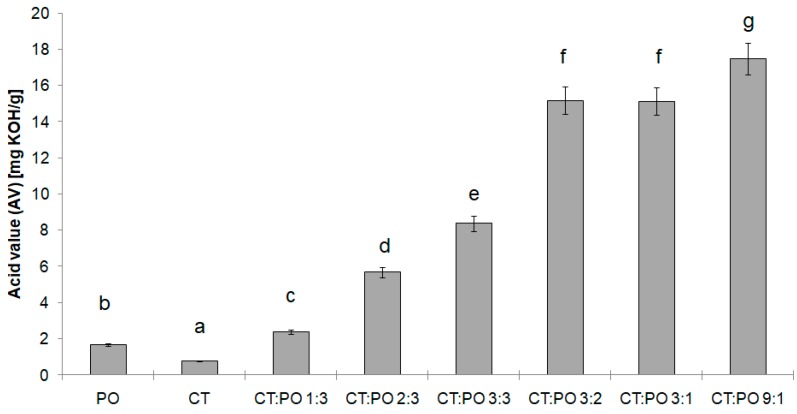
Acid value (AV) of the raw fats and interesterified fat blends. Different letters indicate statistically significant difference (*p* < 0.05).

**Figure 2 biomolecules-10-00115-f002:**
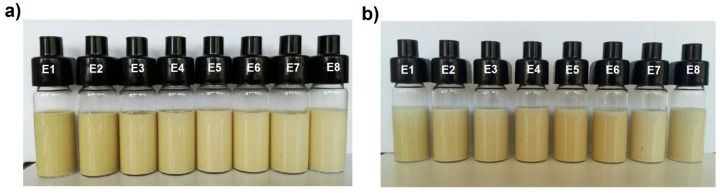
Visual appearance of the emulsions (**a**) after 24 h and (**b**) after 30 days of storage

**Figure 3 biomolecules-10-00115-f003:**
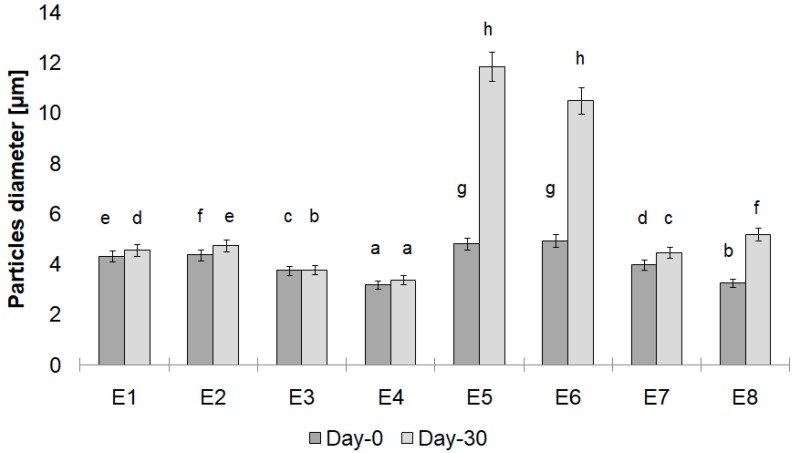
Particle size of emulsions. Day-0: measurement taken on the freshly prepared emulsions. Day-30: measurement taken on the emulsions stored 30 days at 30 °C. Different letters indicate statistically significant difference (*p* < 0.05).

**Figure 4 biomolecules-10-00115-f004:**
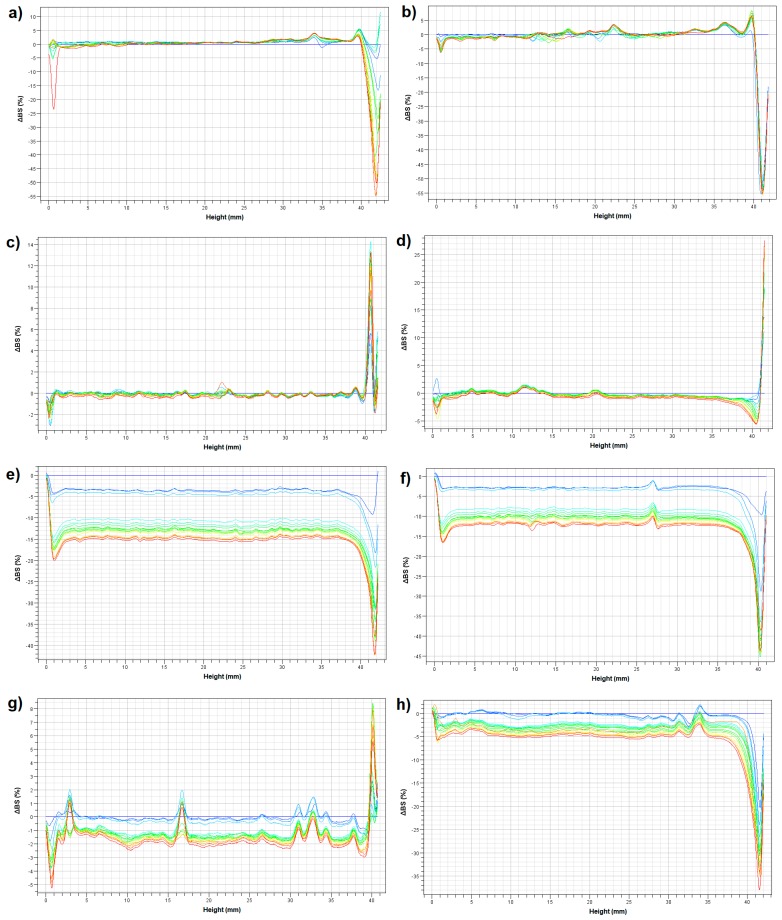
Backscattered light intensity profiles of the prepared emulsions in reference mode. (**a**) E1, (**b**) E2, (**c**) E3, (**d**) E4, (**e**) E5, (**f**) E6, (**g**) E7, (**h**) E8.

**Figure 5 biomolecules-10-00115-f005:**
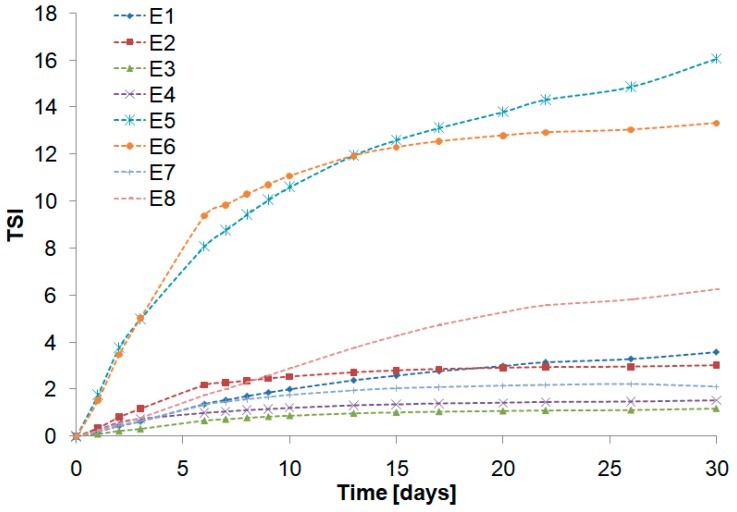
Turbiscan stability index (TSI) variation for the prepared emulsions.

**Figure 6 biomolecules-10-00115-f006:**
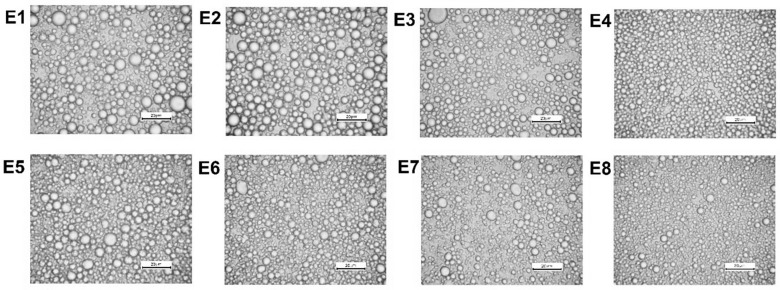
Microphotographs of the freshly prepared emulsions (G× 400)

**Figure 7 biomolecules-10-00115-f007:**
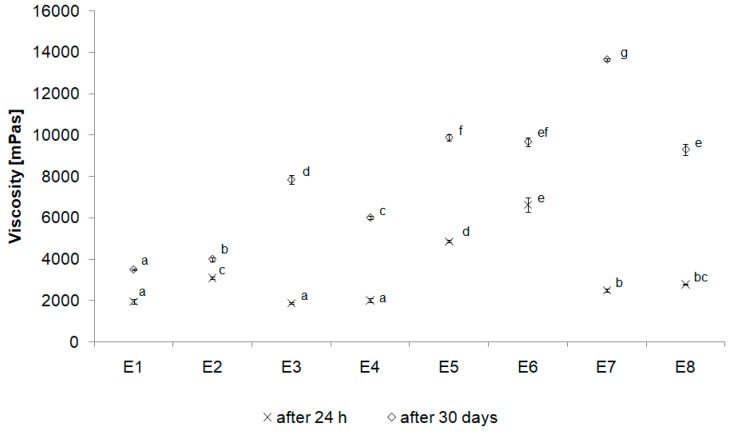
Viscosity of the emulsions after 24 h from their manufacturing and 30 days of storage (as a mean value of 3 determinations ± SD) Different letters indicate statistically significant difference (*p* < 0.05).

**Figure 8 biomolecules-10-00115-f008:**
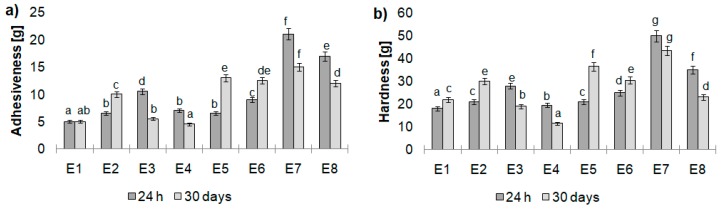
Texture analysis (**a**) adhesiveness and (**b**) hardness of examined emulsions after 24 h and 30 days of storage. Different letters indicate statistically significant difference (*p* < 0.05).

**Table 1 biomolecules-10-00115-t001:** Compositions of the emulsions.

Component	Emulsion							
	E1	E2	E3	E4	E5	E6	E7	E8
Thickener type ^1^	MD	MC & XG	XG	CMC	MD	MC&XG	XG	CMC
Fat type	CIE CT:PO 1:3				CIE CT:PO 3:1			
Thickener wt%	1.0							
CIE blend wt%	30.0							
Lecithin wt%	5.2							
Preservative wt%	0.3							
Water wt%	Up to 100.0							

^1^ MD—maltodextrin, MC & XG—microcrystalline cellulose and xanthan gum, XG—xanthan gum, CMC—carboxymethylcellulose.

**Table 2 biomolecules-10-00115-t002:** Fatty acids profile of the raw fats and interesterified fat blends.

Sample	Fat Type	14:0 (%)	16:0 (%)	16:1 (9-cis) (%)	17:0 (%)	18:0 (%)	18:1 (9-cis) (%)	18:1 (9-trans) (%)	18:2 (all-cis) n-6 (%)	18:2 (all-cis) n-3 (%)	Other (%)
PO	NIE	N/D	14.9 ± 0.5 a	N/D	N/D	5.3 ± 0.2 a	31.9 ± 0.4 a	0.8 ± 0.02 a	46.7 ± 0.5 h	N/D	0.4 ± 0.01 a
CT	NIE	6.3 ± 0.4 g	30.3 ± 0.6 h	3.2 ± 0.3 g	1.1 ± 0.01 g	16.2 ± 0.3 h	35.4 ± 0.3 e	1.8 ± 0.3 f	1.3 ± 0.2 a	0.3 ± 0.01 b	4.1 ± 0.2 f
CT:PO 9:1	CIE	5.7 ± 0.3 f	27.7 ± 0.5 g	2.9 ± 0.2 f	1.0 ± 0.02 f	14.3 ± 0.4 g	36.3 ± 0.5 f	1.8 ± 0.2 f	6.0 ± 0.3 b	0.3 ± 0.01 b	3.8 ± 0.1 ef
CT:PO 3:1	CIE	4.8 ± 0.4 e	26.1 ± 0.4 f	2.5 ± 0.2 e	0.9 ± 0.02 e	13.4 ± 0.3 f	35.2 ± 0.4 e	1.5 ± 0.1 e	12.5 ± 0.2 c	0.3 ± 0.02 b	2.9 ± 0.2 de
CT:PO 3:2	CIE	3.8 ± 0.2 d	23.9 ± 0.5 e	2.0 ± 0.3 d	0.7 ± 0.01 d	12.5 ± 0.3 e	34.3 ± 0.6 d	1.4 ± 0.1 de	18.8 ± 0.3 d	0.3 ± 0.02 b	2.2 ± 0.2 cd
CT:PO 3:3	CIE	3.3 ± 0.3 c	22.7 ± 0.3 d	1.7 ± 0.1 c	0.6 ± 0.02 c	11.2 ± 0.2 d	33.8 ± 0.4 c	1.3 ± 0.2 cd	23.4 ± 0.4 e	0.3 ± 0.01 b	1.8 ± 0.2 bc
CT:PO 2:3	CIE	2.7 ± 0.3 b	21.1 ± 0.2 c	1.5 ± 0.2 b	0.5 ± 0.03 b	10.1 ± 0.3 c	33.2 ± 0.4 b	1.2 ± 0.1 c	28.0 ± 0.4 f	0.2 ± 0.02 a	1.4 ± 0.1 ab
CT:PO 1:3	CIE	1.8 ± 0.2 a	19.2 ± 0.3 b	0.9 ± 0.1 a	0.4 ± 0.01 a	8.7 ± 0.2 b	33.0 ± 0.5 b	1.0 ± 0.03 b	34.5 ± 0.5 g	0.2 ± 0.01 a	0.2 ± 0.01 a

PO—pumpkin seed oil; CT—calf tallow; CIE—chemically interesterified; NIE—non-interesterified; N/D—not detected; Means in the same column with different letters differ significantly (*p* < 0.05).

**Table 3 biomolecules-10-00115-t003:** Polar and nonpolar fraction content of the raw fats and interesterified blends.

Fat Type		TAG (%)	DAG (%)	FFA (%)	MAG (%)
**Raw fats**	**PO**	97.7 ± 0.7 e	1.5 ± 0.03 a	0.8 ± 0.02 b	N/D
	**CT**	99.5 ± 0.6 f	N/D	0.4 ± 0.03 a	0.1 ± 0.01 a
**CIE blends**	**CT:PO 9:1**	82.2 ± 0.6 a	6.6 ± 0.04 c	8.8 ± 0.06 g	2.4 ± 0.02 d
	**CT:PO 3:1**	84.5 ± 0.8 bc	7.4 ± 0.06 d	7.6 ± 0.06 f	0.5 ± 0.01 c
	**CT:PO 3:2**	83.6 ± 0.7 b	8.4 ± 0.04 g	7.6 ± 0.05 f	0.3 ± 0.02 b
	**CT:PO 3:3**	85.5 ± 0.7 c	8.0 ± 0.05 f	4.2 ± 0.04 e	2.3 ± 0.03 d
	**CT:PO 2:3**	85.0 ± 0.6 c	7.7 ± 0.06 e	2.9 ± 0.03 d	4.5 ± 0.03 f
	**CT:PO 1:3**	89.5 ± 0.9 d	6.2 ± 0.06 b	1.2 ± 0.03 c	3.1 ± 0.02 e

PO—pumpkin seed oil; CT—calf tallow; CIE—chemically interesterified; NIE—non-interesterified; N/D—not detected; TAG—triacylglycerol; DAG—diacylglycerol; MAG—monoacylglycerol; FFA—free fatty acids; Means in the same column with different letters differ significantly (*p* < 0.05).

**Table 4 biomolecules-10-00115-t004:** CIELAB L*, a*, b* values of the emulsions stored 30 days at 5 °C (as a mean value of 3 determinations ± SD).

Time	48h			1 month		
Emulsion	L*	a*	b*	L*	a*	b*
**E1**	12.09 ± 0.08 b	−0.43 ± 0.02 a	9.19 ± 0.14 b	7.42 ± 0.44 b	−0.52 ± 0.17 a	6.47 ± 0.21 a
**E2**	11.95 ± 0.06 a	−0.49 ± 0.06 c	9.88 ± 0.08 c	7.15 ± 0.13 a	−0.81 ± 0.13 c	6.53 ± 0.16 b
**E3**	12.43 ± 0.06 c	−0.70 ± 0.05 e	9.03 ± 0.02 a	8.02 ± 0.25 c	−0.78 ± 0.10 b	7.09 ± 0.17 c
**E4**	12.97 ± 0.03 d	−0.46 ± 0.04 b	9.88 ± 0.02 c	8.08 ± 0.39 d	−0.80 ± 0.05 bc	7.13 ± 0.06 c
**E5**	13.38 ± 0.02 e	−0.66 ± 0.07 d	10.24 ± 0.06 e	8.99 ± 0.37 e	−1.21 ± 0.22 g	7.86 ± 0.25 e
**E6**	13.01 ± 0.03 d	−0.51 ± 0.11 c	10.87 ± 0.05 g	9.27 ± 0.41 f	−0.87 ± 0.07 d	8.05 ± 0.24 f
**E7**	13.75 ± 0.03 f	−0.77 ± 0.06 f	10.13 ± 0.09 d	8.10 ± 0.59 d	−0.91 ± 0.09 e	7.42 ± 0.48 d
**E8**	14.27 ± 0.01 g	−0.78 ± 0.03 f	10.32 ± 0.04 f	10.63 ± 0.19 g	−0.94 ± 0.03 f	8.37 ± 0.19 g

Means in the same column with different letters differ significantly (*p* < 0.05).

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
