# Peer review of "Physicochemical Characterization and Evaluation of Emulsions Containing Chemically Modified Fats and Different Hydrocolloids"

_biomolecules, 2020, doi:10.3390/biom10010115_

Round 1

Reviewer 1 Report

In the following I report some key points that in my opinion should be addressed by the authors before the paper would be accepted for the publications: 

Why did the authors choose pumpkin seed oil and calf tallow for this research? How  have the authors determined the content of lecithin or other ingredients such as fat in prepared emulsions? The authors should refer to the observed trend that the acid number of fat mixtures increased with increasing solid fat in fat blend. How do the authors explain this? Why two fats with the composition (3: 1) (1: 3) were chosen for further research to manufacture the emulsions? In the manuscript there is a lack of description : how the particle size of emulsion was determined? In my opinion, the manuscript title lacks information on factors that were also taken into account. I propose a possible following title: Physicochemical characterization and evaluation of emulsions containing chemically modified fats and different hydrocolloids. No statistical analysis on results is available in the paper – please supplement it.

Author Response

Radom, January 03, 2020

Article title:

Physicochemical characterization and evaluation of emulsions containing structured fats formed by chemical interesterification

Małgorzata Kowalska , Magdalena Woźniak, Anna Żbikowska, Mariola Kozłowska

Manuscript ID 686928

Dear Editor,

Biomolecules,

The authors would like to thank the reviewers for all kind comments in the reviews. All reviewers' suggestions were taken into account and the text of the manuscript was corrected. Certainly, this will affect the quality of the paper and make it more clear (accessible) to the reader. All corrections were introduced into the manuscript in red color. The detailed answers to reviewers’ queries are placed below.

Reviewer(s)' Comments to Author:

Reviewer: 1

Comments to the Author

Why did the authors choose pumpkin seed oil and calf tallow for this research?

Beef tallow is a by-product of a meat industry, and is considered as not suitable for direct human consumption due to low nutritional value (low content of polyunsaturated fatty acids). However, this fat has beneficial properties, such as high thermal and oxidative stability, or favorable plasticity at room temperature. Triacylglycerols of this fat contain a relatively high content of saturated fatty acids, which can cause a sandy sensation. Under conditions of frequent temperature changes (e.g. during storage or transport), the crystal structure of this fat may change, leading to the formation of crystals up to 2-3 mm in size. In order to increase its nutritional value and potential uses, it is subjected to modifications with various vegetable oils. The choice of pumpkin seed oil in our research was dictated by, among others high content of bioactive compounds as tocopherols, sterols, β-carotene, and lutein. The highly unsaturated fatty acid composition of pumpkin seed oil makes it well-suited for improving nutritional benefits. Due to the characteristic nutty taste as well as a deep color, in the authors' opinion it could be a novelty for model fat emulsions.

The justification of the fats choice were implemented in the manuscript.

How have the authors determined the content of lecithin or other ingredients such as fat in prepared emulsions?

Our previous studies concerned optimization of the composition of new components introduced into the emulsion. Development of the emulsion composition was published in the following publications:

Kowalska, M., Ziomek, M., & Żbikowska, A. (2015). Stability of cosmetic emulsion containing different amount of hemp oil. International Journal of Cosmetic Science, 37(4), 408-416. Kowalska M., Żbikowska A., Śmiechowski K., Marciniak-Łukasiak K.: Wpływ ilości lecytyny słonecznikowej i czasu homogenizacji na stabilność emulsji spożywczej zawierającej olej z orzechów włoskich. Żywność. Nauka. Technologia. Jakość. 2014, 1(92): 78-91, (manuscript was published in Polish; title, abstract, key words, figures and tables captions are available in English)

The authors should refer to the observed trend that the acid number of fat mixtures increased with increasing solid fat in fat blend. How do the authors explain this?

The work actually shows a trend of increasing acid value for the blends with increasing animal fat content. It testified that the hydrolysis process for these blends was more advanced when the triacylglycerol animal fat share was greater. The discussion was provided.

Why two fats with the composition (3:1) (1:3) were chosen for further research to manufacture the emulsions?

The choice of these fat mixtures was dictated by different plasticity and consistency, which, according to the authors, could also have an impact on their physicochemical stability.

(Information implemented in the text)

In the manuscript there is a lack of description : how the particle size of emulsion was determined?

The fragment concerning the particle size determination was separated.

In my opinion, the manuscript title lacks information on factors that were also taken into account. I propose a possible following title:Physicochemical characterization and evaluation of emulsions containing chemically modified fats and different hydrocolloids. 

The title was modified.

No statistical analysis on results is available in the paper – please supplement it. 

Statistical analysis was supplemented. Minor errors were also corrected.

Reviewer 2 Report

The manuscript “Physicochemical characterization and evaluation of emulsions containing structured fats formed by chemical interesterification” is interesting since the use of interesterified lipids from animal (calf tallow) and vegetal (pumpkin oil) sources is proposed for the preparation of emulsions. Despite in the introduction several improved properties of the interesterified lipids are claimed, it is not clear from the manuscript which is the real advantage of preparing and using in the emulsion formulation these modified lipids. For instance, in the introduction the improved nutritional properties are mentioned or also solid fat content but this aspects have not been evaluated experimentally in the produced mixtures. I suggest revising the entire introduction in order to be more straightforward and concise on the real aim of the work.

Please define CIE (Line 50 ) at the first appearance

In the material section, please add the country of each provider

I have some concerns regarding emulsion preparation. How the percentage of each component was selected? Please add a reference. Why emulsions were prepared only with CT:PO 1:3 and CT:PO 3:1 ratio interesterified lipids? Was enough only 1 minute for dispersing the thickening agents since most of them are hydrocolloids? In the manuscript it is widely discussed the role of the thickening agent for stabilizing the emulsion, but there is no mention regarding the surfactant. The only surfactant in the formulation is lecithin, please discuss about that. Please add the rpm used for the homogenization by ultra-turrax. Emulsions should also be prepared with not processed pumpkin oil as reference and to compare their chemical-physical properties with those prepared with the interesterified mixtures.

It is also not clear if the obtained interesterified lipid are liquid or solid at room temperature.

Line 267 cream layer. Do you mean that no creaming was observed for all prepared emulsions after 30 days of storage?

Please in the conclusions focus more on the achievement of the study

Author Response

Radom, January 03, 2020

Article title:

Physicochemical characterization and evaluation of emulsions containing structured fats formed by chemical interesterification

Małgorzata Kowalska , Magdalena Woźniak, Anna Żbikowska, Mariola Kozłowska

Manuscript ID 686928

Dear Editor,

Biomolecules,

The authors would like to thank the reviewers for all kind comments in the reviews. All reviewers' suggestions were taken into account and the text of the manuscript was corrected. Certainly, this will affect the quality of the paper and make it more clear (accessible) to the reader. All corrections were introduced into the manuscript in red color. The detailed answers to reviewers’ queries are placed below.

Reviewer(s)' Comments to Author:

Reviewer: 2

The manuscript “Physicochemical characterization and evaluation of emulsions containing structured fats formed by chemical interesterification” is interesting since the use of interesterified lipids from animal (calf tallow) and vegetal (pumpkin oil) sources is proposed for the preparation of emulsions. Despite in the introduction several improved properties of the interesterified lipids are claimed, it is not clear from the manuscript which is the real advantage of preparing and using in the emulsion formulation these modified lipids. For instance, in the introduction the improved nutritional properties are mentioned or also solid fat content but this aspects have not been evaluated experimentally in the produced mixtures. I suggest revising the entire introduction in order to be more straightforward and concise on the real aim of the work.

The purpose of this work was an application of chemically interesterified fats in emulsions. It is well known that interesterification has a positive effect on the physical properties of fats subjected to this process. Among others it affects the change in the crystal structure, and thus causes a smoothing effect. Due to this fact, it seemed interesting to use this type of chemically modified fat in emulsions, because, to the best of our knowledge, very few authors have conducted research in this direction. In our opinion, in a way, the study of nutritional value was presented in the work by analyzing the composition of fatty acids. By using waste animal fat, which was enriched with unsaturated fatty acids derived from pumpkin seed oil, its nutritional value was increased.

As a consequence, it was only in our work that we managed to indicate a new direction in the application of chemically interesterified fats and to confirm that it is possible to produce model dispersion systems based on new fats with more favorable physicochemical properties than their original raw materials.

(Required information was added into the introduction)

Please define CIE (Line 50 ) at the first appearance

The remark was implemented.

In the material section, please add the country of each provider

The country of each provider was added.

I have some concerns regarding emulsion preparation. How the percentage of each component was selected? Please add a reference.

Percentage of each component was based on our previous studies. The references were added to the manuscript.

Kowalska M., Żbikowska A., Śmiechowski K., Marciniak-Łukasiak K.: Wpływ ilości lecytyny słonecznikowej i czasu homogenizacji na stabilność emulsji spożywczej zawierającej olej z orzechów włoskich. Żywność. Nauka. Technologia. Jakość. 2014, 1(92): 78-91, (manuscript was published in Polish; abstract, key words, figures and tables captions are available in English) Kowalska, M., Ziomek, M., Żbikowska, A. (2015). Stability of cosmetic emulsion containing different amount of hemp oil. International journal of cosmetic science, 37(4), 408-416.)

Why emulsions were prepared only with CT:PO 1:3 and CT:PO 3:1 ratio interesterified lipids?

In our opinion, based on previous studies, the most interesting was a comparison of at least 2 extreme compositions, hence the choice of 3:1 and 1:3 fat ratio. ( An explanation was introduced into the text)

Was enough only 1 minute for dispersing the thickening agents since most of them are hydrocolloids?

Yes, it was enough. The experiment was conducted on blank samples (thickener dispersion in water) at different homogenization times (30, 60, 90 seconds) and it turned out that 1 minute is sufficient time for the thickener to be properly dispersed. (data not shown - auxiliary data)

In the manuscript it is widely discussed the role of the thickening agent for stabilizing the emulsion, but there is no mention regarding the surfactant. The only surfactant in the formulation is lecithin, please discuss about that.

Information on lecithin has been added to the manuscript.

Please add the rpm used for the homogenization by ultra-turrax.

Information about rpm used was added.

Emulsions should also be prepared with not processed pumpkin oil as reference and to compare their chemical-physical properties with those prepared with the interesterified mixtures.

We agree with this suggestion, although the presented research was directed to the possibility of using chemically interesterified fat blends to obtain stable emulsion systems. When assessing the effect of interesterification of a fatty base on emulsion stability and properties, such a comparison would definitely be necessary.

It is also not clear if the obtained interesterified lipid are liquid or solid at room temperature.

Interesterified CT:PO 1:3 blend is semi-solid at room temperature, and CT:PO 3:1 blend is solid.

Line 267 cream layer. Do you mean that no creaming was observed for all prepared emulsions after 30 days of storage?

Yes. It is.

Please in the conclusions focus more on the achievement of the study

The conclusions were corrected.

Reviewer 3 Report

This manuscript studied the effect of chemical transesterification on the properties of lipids and emulsion products, the content of this study is innovative, however, there are some problems in this manuscript.

Why are different emulsifiers chosen? Please provide the basis for your choice. Please explain the reasons for choosing the single emulsifier and compound emulsifiers. The preparation of emulsion lacks homogenization process and the nature of the crude emulsion is unstable. With regard to emulsion products, what is the concentration of emulsifiers E2 and E6? The samples should contain controls of the original DO and CT lipids. There is no statistical analysis both in the method and the results. The unit (%) in Table 2 should be supplemented. What do TAG, DAG, FFA, MAG mean in Table 3? Legend should be supplemented at the bottom of the table. With regard to particle size results, samples E7>E6, E3>E2, why? The effect of composition of lipids on the properties of emulsions were scarcely discussed, and more literatures should be cited. The application of the emulsion to the skin is far-fetched because of the lack of padding in the previous discussion.

Author Response

Radom, January 03, 2020

Article title:

Physicochemical characterization and evaluation of emulsions containing structured fats formed by chemical interesterification

Małgorzata Kowalska , Magdalena Woźniak, Anna Żbikowska, Mariola Kozłowska

Manuscript ID 686928

Dear Editor,

Biomolecules,

The authors would like to thank the reviewers for all kind comments in the reviews. All reviewers' suggestions were taken into account and the text of the manuscript was corrected. Certainly, this will affect the quality of the paper and make it more clear (accessible) to the reader. All corrections were introduced into the manuscript in red color. The detailed answers to reviewers’ queries are placed below.

Reviewer(s)' Comments to Author:

Reviewer: 3

This manuscript studied the effect of chemical transesterification on the properties of lipids and emulsion products, the content of this study is innovative, however, there are some problems in this manuscript. Why are different emulsifiers chosen? Please provide the basis for your choice. Please explain the reasons for choosing the single emulsifier and compound emulsifiers.

There was one emulsifier used – lecithin – the information about this choice was added in the manuscript. Various thickeners were applied. The choice was dictated by their properties.

We wanted to use thickeners that are safe and widely used in food and cosmetic emulsions. The decision was also made on the basis of our previous experience (mentioned below), which were published in Polish-language journals, hence due to limited access they were not included as references, although information explaining the selection of a thickeners was introduced in the manuscript.

Kowalska M., Żbikowska A., Górecka A.: Wpływ wybranych zagęstników na rozkład kropel oleju w emulsjach niskotłuszczowych. Żywność. Nauka. Technologia. Jakość. 2011, 4(77): 84-93 Kowalska M., Górecka A., Śmiechowski K., Krygier K.: Fizyczna stabilność emulsji niskotłuszczowej w zależności od zastosowanych hydrokoloidów. Żywność. Nauka. Technologia. Jakość. 2006, 2(47),143 -152 Kowalska M, Żbikowska A. Górecka A.: Badania stabilności emulsji spożywczych z dodatkami różnych zagęstników. Bromatologia i Chemia Toksykologiczna. 2011, 44(3): 883-889,

Our goal was also testing single thickener, as well as determination of the synergism of the action of two associated thickeners. For this purpose we used one ready-to-use preparation consisting of 2 compounds (Microcrystalline Cellulose and Xanthan Gum). It is a commercial product of J. Rettenmaier & Söhne, used, among others, in commercial emulsion products.

The preparation of emulsion lacks homogenization process and the nature of the crude emulsion is unstable.

Homogenization of the emulsion was carried out using the Ultra-Turrax mechanical homogenizer, the process description was slightly modified in section 2.3.1.

With regard to emulsion products, what is the concentration of emulsifiers E2 and E6?

We do not have this information, it is a commercial product. Information which we do have is company's own recommendations as to the amount of use and properties of the product.

The samples should contain controls of the original DO and CT lipids.

We agree with this suggestion, when assessing the effect of fatty base interesterification on emulsion stability and properties, such a comparison would be necessary. In the presented work, we focused rather on the possibility of using chemically esterified fat blends to obtain stable emulsion systems.

There is no statistical analysis both in the method and the results.

Statistical analysis was supplemented. Minor errors were also corrected.

The unit (%) in Table 2 should be supplemented.

The “%” was supplemented.

What do TAG, DAG, FFA, MAG mean in Table 3? Legend should be supplemented at the bottom of the table.

The legend was supplemented.

With regard to particle size results, samples E7>E6, E3>E2, why?

The use of different types of thickener, even more in different amounts, can result in various average particle size of the emulsion. We have also conducted such tests and confirmed that the type of thickener affects the size of the emulsion droplets.

Kowalska M., Żbikowska A., Górecka A.: Wpływ wybranych zagęstników na rozkład kropel oleju w emulsjach niskotłuszczowych. Żywność. Nauka. Technologia. Jakość. 2011, 4(77): 84-93,

The effect of composition of lipids on the properties of emulsions were scarcely discussed, and more literatures should be cited.

Corresponding changes have been implemented in the text. Many authors have conducted research in the field of chemically interesterified fats - mainly physical and chemical changes that occur in them, but we have not found specific studies regarding the production of fats and their application in emulsion systems. Therefore, the assessment of the emulsions based on interesterified fats conducted by other authors was difficult for us to find. Of course, the parameters and behavior of the emulsion are described during a given storage period, but there are no data on similar systems that we dealt with.

The application of the emulsion to the skin is far-fetched because of the lack of padding in the previous discussion. 

In the authors' opinion, application of this kind of products on the skin is possible and even beneficial. Based on our previous research (Kowalska M., Woźniak M., Mróz D., Janas S., 2018. A comparison of physicochemical properties of an emulsion containing chemically interesterified fat for demanding skin with commercial formulations for atopic skin. Journal of Cosmetic Sciences. 69, 6: 411-428), an emulsion containing chemically interesterified mutton tallow and sesame oil have successfully reduced transepidermal water loss.

Another study (Kowalska M., Mendrycka M., Żbikowska A., Stawarz S.: Enzymatically interesteried fats based on mutton tallow and walnut oil suitable for cosmetic emulsion. International Journal of Cosmetic Sciences. 2015, 37(1): 82-91) has shown that emulsions containing interesteried fats were acceptable by the respondents, and revealed favorable sensory properties.

Reviewer 4 Report

The overall quality of the manuscript is good. The article is prepared in accordance with the editorial guidelines of Biomolecules

Title clearly describes what the manuscript is about. Abstract adequately describes the study, principle results and conclusions. Employed experimental methods are adequate, sufficiently clear and complete to allow repetition of the work. Data are properly analyzed and interpreted to support the conclusions. Pictures and Table are satisfactory and interpreted correctly. Relevant issues in discussion are adequately discussed. Cited references are appropriate.

The present study (under my review)  sought to develop structured lipids based on pumpkin seed oil and calf tallow blends at various ratios aiming to obtain a product with better physical and chemical properties and desirable functionality to be applied in many industries. In my opinion its look great.

The idea of connection pumpkin seed oil and calf tallow blends in one emulsion gives a picture of the great scientific commitment and cognitive skills of the scientific team. Each approach and problem solution is important and noteworthy. The presented research results are extremely valuable and promising. Congratulations on the idea and approach.

As an additional "plus" it should be noted that the Research Team has a lot of experience and scientific achievements, which significantly increases the range of this publication. This probably also had a big impact on the use of many interesting and modern analytical techniques in this study.

I think this is a very valuable manuscript.

In my opinion… the article is suitable for publication in the journal Biomolecules.

Author Response

Radom, January 03, 2020

Article title:

Physicochemical characterization and evaluation of emulsions containing structured fats formed by chemical interesterification

Małgorzata Kowalska , Magdalena Woźniak, Anna Żbikowska, Mariola Kozłowska

Manuscript ID 686928

Dear Editor,

Biomolecules,

The authors would like to thank the reviewers for all kind comments in the reviews. All reviewers' suggestions were taken into account and the text of the manuscript was corrected. Certainly, this will affect the quality of the paper and make it more clear (accessible) to the reader. All corrections were introduced into the manuscript in red color. The detailed answers to reviewers’ queries are placed below.

Reviewer(s)' Comments to Author:

Reviewer 4

The overall quality of the manuscript is good. The article is prepared in accordance with the editorial guidelines of Biomolecules Title clearly describes what the manuscript is about. Abstract adequately describes the study, principle results and conclusions. Employed experimental methods are adequate, sufficiently clear and complete to allow repetition of the work. Data are properly analyzed and interpreted to support the conclusions. Pictures and Table are satisfactory and interpreted correctly. Relevant issues in discussion are adequately discussed. Cited references are appropriate. The present study (under my review)  sought to develop structured lipids based on pumpkin seed oil and calf tallow blends at various ratios aiming to obtain a product with better physical and chemical properties and desirable functionality to be applied in many industries. In my opinion its look great. The idea of connection pumpkin seed oil and calf tallow blends in one emulsion gives a picture of the great scientific commitment and cognitive skills of the scientific team. Each approach and problem solution is important and noteworthy. The presented research results are extremely valuable and promising. Congratulations on the idea and approach. As an additional "plus" it should be noted that the Research Team has a lot of experience and scientific achievements, which significantly increases the range of this publication. This probably also had a big impact on the use of many interesting and modern analytical techniques in this study. I think this is a very valuable manuscript. In my opinion… the article is suitable for publication in the journal Biomolecules.

Thank you very much for the positive assessment of our work.

Round 2

Reviewer 2 Report

The authors have addressed the reviewer's coomments properly. The manuscript is suitable for pubblication

Reviewer 3 Report

The author has made careful revisions and the paper is ready for publication.